# Factors Associated with Behavioral and Psychological Symptoms of Dementia during COVID-19

**DOI:** 10.3390/ijerph191610094

**Published:** 2022-08-15

**Authors:** Yujiro Kuroda, Taiki Sugimoto, Kenichi Satoh, Claudia K. Suemoto, Nanae Matsumoto, Kazuaki Uchida, Yoshinobu Kishino, Takashi Sakurai

**Affiliations:** 1Department of Prevention and Care Science, Center for Development of Advanced Medicine for Dementia, National Center for Geriatrics and Gerontology, Obu 474-8511, Japan; 2Faculty of Data Science, Shiga University, Hikone 522-8522, Japan; 3Division of Geriatrics, University of Sao Paulo, Sao Paulo 01246-903, Brazil; 4Center for Comprehensive Care and Research on Memory Disorders, National Center for Geriatrics and Gerontology, Obu 474-8511, Japan; 5Department of Cognition and Behavior Science, Nagoya University Graduate School of Medicine, Nagoya 464-8601, Japan

**Keywords:** behavioral and psychological symptoms, social distancing, dementia, coronavirus disease, mild cognitive impairment

## Abstract

(1) Background: Preventive measures to control the spread of COVID-19 are essential, but they often cause social isolation and diminish the physical and mental health of older adults. In cognitively impaired individuals, the pandemic has worsened behavioral and psychological symptoms of dementia (BPSD). Here, we explored the factors contributing to the worsening of BPSD during the COVID-19 pandemic. (2) Methods: Potential patients were identified at a memory clinic in Japan between June 2017 and June 2021. Eligible patients had a diagnosis of mild cognitive impairment (MCI) or dementia during the study period. The outcome was BPSD, as assessed by using the Dementia Behavioral Disorders Scale. Information on patients’ lifestyle habits and use of care services was obtained for use as primary explanatory variables; multiple regression analysis was performed to examine the relationship between BPSD and care services use or lifestyle habits. The model was adjusted for sociodemographic factors, and the interaction terms of the pandemic period with lifestyle and service use were included to evaluate the effects of COVID-19. (3) Results: We identified 977 participants with MCI and 1380 with dementia (MCI group: 69.8% age 75 years or older, 54.2% female; dementia group: 79.8% age 75 years or older, 64.8% female). After adjustment for possible confounders, significantly worse BPSD was demonstrated in those who used daycare services during COVID-19 (both MCI and dementia patients; *p* = 0.007 and *p* = 0.025 respectively) and in those with poor nutritional function (dementia patients; *p* = 0.040). (4) Conclusions and Implications: During COVID-19, poor nutritional status and use of daycare services were associated with BPSD in those with cognitive decline. These findings indicate the need to fully examine the quantity and quality of care services for people with cognitive decline during emergencies and to continue to provide effective services.

## 1. Introduction

Although it is essential to implement measures to limit the spread of COVID-19, such measures often lead to social isolation, which can negatively influence the mental and physical health of older adults. The restrictions imposed because of the pandemic are likely to have worsened the behavioral and psychological symptoms of dementia (BPSD) in patients with dementia or other forms of cognitive decline [1].

Confinement and isolation are effective ways of controlling COVID-19 and other viral infections, as shown by systematic reviews [2]. However, studies of past outbreaks (e.g., of MERS and SARS) have found that long-term “lockdowns” or quarantining as public health measures can compromise mental health and increase psychological signs and symptoms of the stress response, including depression and anxiety [3,4]. However, these studies examined general populations, and we have very little information on socially vulnerable older adults or people with dementia [1,5]. Those with dementia are frail; they frequently depend on caregivers [6] and often require nursing care and other social services [7,8]. If they are in long-term isolation, they cannot access social services, such as daycare and dementia cafés, and this inability can negatively influence their behavior (i.e., BPSD), placing an increased burden on their caregivers [9].

The causes and risk factors for BPSD are diverse and include biological, psychological, and environmental variables [10]. Often, their combination, rather than any specific factor, explains the occurrence of BPSD. Furthermore, protective factors include the person’s lifestyle and the use of care services [11,12]. However, with the outbreak of the COVID-19 pandemic, precautionary measures were taken to limit the number of gatherings in Japan, and the use of outpatient and long-term care services significantly decreased throughout the country [13]. Furthermore, the services provided in long-term care facilities were also affected by the precautionary measures, which included social distancing and other infection-control measures [14]. However, despite these preventive measures, COVID-19 infection clusters still formed in nursing homes, and services had to be suspended during specific periods. Thus, the reduction or suspension of services due to the pandemic may have worsened the dementia symptoms of patients, consequently increasing the caregiving burden and psychological and physical health problems among care professionals.

BPSD may have worsened in dementia patients, or newly developed in patients with mild cognitive impairment (MCI), during the COVID-19 pandemic [15,16,17]. An Italian phone survey of patients with dementia or mild cognitive impairment found that 32% showed memory and orientation deterioration; 8% had a functional decline in activities of daily living (ADL) (e.g., reduced independence in housekeeping and personal care) [15]. In about 60% of the patients, BPSD (apathy, agitation or aggression, depression) either newly developed or worsened under the stress of COVID-19, increasing the neuropsychiatric burden on these patients [17].

These studies were conducted only after the start of the COVID-19 pandemic, and the causality of the increase in BPSD is not clear. Therefore, in our previous study, we examined the prevalence of BPSD using continuous data from 15 months before and during the COVID-19 pandemic and found that sleep disturbances and aggressiveness were more prevalent after the start of COVID-19 [18]. Considering the results of previous studies [15,17], we can conclude that some cases of BPSD have worsened since the start of the COVID-19 pandemic, but the factors that have contributed to this deterioration are unclear. Our next step was to explore the factors associated with BPSD during the COVID-19 pandemic to address clinical demands in patients with cognitive impairment. We hypothesized that the existing protective factors (i.e., care services [11]) were not functionally effective during COVID-19 and that individuals’ lifestyles had changed [12], making BPSD more likely to emerge. Therefore, this study focused on the impacts of care services and lifestyle on the prevalence of BPSD before and during COVID-19.

## 2. Materials and Methods

### 2.1. Study Cohort

The aim of this study was to analyze BPSD factors during two different periods, before and during the COVID-19 pandemic. Therefore, a cross-sectional study was conducted at two different time points. Participants were patients at a memory clinic in Aichi, Japan, from June 2017 to December 2019 (a 30-month period before the onset of the COVID-19 pandemic; “before COVID-19”, *n* = 1931), or from April 2020 to June 2021 (a 15-month period after the declaration of the COVID-19 emergency; “during COVID-19”, *n* = 619 patients). Assessments were conducted for first-time outpatients during both of the study periods. Patients had a dementia-related diagnosis according to the criteria of the National Institute on Aging–Alzheimer’s Association workgroups [19,20]. Specifically, MCI due to Alzheimer’s disease [19] or dementia was classified as either probable or possible AD [20], probable or possible dementia with Lewy bodies and Parkinson’s disease [21,22], or vascular dementia [23]. We selected 1781 before-COVID-19 and 576 during-COVID-19 participants after excluding those with missing variables, including the primary outcome measure of the study, a score on the Dementia Behavioral Disturbance (DBD) scale [24]. The dataset used in this study was designed to be highly extensive before the COVID-19 period compared to the dataset used in our previous study [18], resulting in an increased number of participants. Independent researchers ensured that the same participants were not evaluated at the two time points. The study protocol was approved by an institutional ethics committee.

### 2.2. Outcomes

BPSD was assessed with the DBD [24], which consists of 28 observable behaviors related to dementia, such as passivity, agitation, eating disturbances, aggressiveness, diurnal rhythm disturbances, and sexual disinhibition. The frequency of each item is rated by a primary caregiver on a scale of 0 to 4 (0 = never, 1 = rarely, 2 = sometimes, 3 = frequently, 4 = always), with higher scores indicating greater severity of BPSD. The Japanese version of the DBD has been validated [25]. The DBD was completed by the primary caregiver (patient’s spouse or child, 90.5%) independent of the patient, and the results were collected by an outpatient health care provider.

### 2.3. Explanatory Variables

Two main explanatory variables were used: (a) the use of care services and (b) the lifestyle of the patient. With regard to the use of care services, the primary caregivers reported the use of six types of care services under the Japanese long-term care insurance system: in-home care, senior daycare center, day healthcare center, short-term admission, assisted living facility, and group home for older adults with dementia [26]. From these, three categories were defined: “Home visiting services (in-home care)”, “Daycare services (senior daycare center, day healthcare center)”, and “Residential services (short-term admission, assisted living facility, and group home for older adults with dementia).” Patient lifestyle was assessed by the primary caregiver on a 4-point scale that evaluated light exercise/physical training, regular smoking, sleep quality, and weight loss. For the “light exercise/physical training” and “sleep quality” variables, a “good” status was counted as 3 or 4 on a 4-point scale, whereas weight loss was counted as “not available (N/A)”, and smoking was counted as “no smoking”. Nutritional status was assessed by using the Mini-Nutrition Assessment–Short Form (MNA-SF) [27]. MNA-SF scores were classified into two groups by using cut-off values: ≤11 points indicated deterioration of nutritional status, as in previous studies [28]. For the regular smoking variable, non-smoking status was counted, and for the weight loss variable, responses where there was a loss were counted.

### 2.4. Other Variables

Cognitive function in older adults was assessed by use of the Mini-Mental State Examination [29]. Scores range from 0 to 30, with higher scores reflecting a higher level of global cognitive performance. Other information, such as gender, age, education, living environment, comorbidity (diabetes mellitus, hypertension, dyslipidemia, cardiac disease, and stroke), and polypharmacy (five or more prescribed medications) [30], was obtained from medical records. In addition, the primary caregiver reported the basic ADL, as assessed by using the Barthel Index [31], and the Instrumental ADL, as assessed by using the Lawton Index [32], with higher scores indicating better status.

To further analyze whether the use of care services (Home visiting, Daycare, or Residential) changed between before and during COVID-19, we obtained data from the National Health Service and conducted a descriptive analysis. Specifically, we presented the monthly trends of use of the three care services indicated above, compared with the same month in the previous year. The data covered the period from April 2010 to October 2021.

### 2.5. Statistical Analysis

We calculated means, standard deviations (SD), frequencies, and percentages to describe the demographic data in the MCI and dementia patients before and during COVID-19. The relationships between the DBD score and care services or lifestyle-related variables were further evaluated by using a multivariate analysis, which was adjusted for three sets of variables: (a) socioeconomic factors (age, gender, education, living with family); (b) physical functioning and use of medical and nursing care services (medical condition, Instrumental ADL score, polypharmacy); and (c) cognitive functioning (Mini-Mental State Exam score). In addition, interaction terms of the COVID-19 period (before = 0; during = 1) were created with each of the care services and lifestyle variables and included in the multivariate analysis model to evaluate the effects of COVID-19. Furthermore, independent-sample *t*-tests were conducted on the Barthel Index to compare physical function before and during COVID-19 in participants who reported using care services (Home visiting, Daycare, or Residential). All analyses were performed by using R (R Foundation for Statistical Computing, Vienna Austria) and STATA v. 16.1 (Stata Corp, College Station, TX, United States). *p*-values < 0.05 were considered statistically significant.

## 3. Results

Within the study period, 1380 patients diagnosed with dementia and 977 individuals with MCI were identified; 79.8% of the dementia patients were age 75 years or older (64.8% female), and 69.8% of the MCI patients were age 75 years or older (54.2% female). In participants with MCI or dementia, there were no significant differences in the use of any of the care services before and during COVID-19. After the start of the pandemic, MCI patients had significantly lower MMSE scores than before the pandemic (*p* = 0.001) (Table 1). The presence of BPSD was defined by responses of “sometimes,” “often,” or “always,” and the prevalence before and during the pandemic were analyzed using the chi-squared test stratified by MCI and Dementia patients (Appendix A).

Multiple regression analysis revealed no main effect of the COVID-19 period (MCI group β = 2.12, CI −4.39 to 8.63; Dementia group β = 1.45, CI −5.74 to 8.64). However, in MCI patients, better quality of sleep (*p* < 0.001) and higher MNA score (*p* = 0.003) were negatively associated with DBD score. In dementia patients, receiving Residential care services (*p* = 0.011), better quality of sleep (*p* = 0.016), higher MNA score (*p* = 0.040), and not smoking (*p* = 0.004) were negatively associated with DBD score, even after adjustment for possible confounders (Table 2). The interaction term between receiving Daycare and period (during COVID-19) was positively associated with DBD score in both MCI patients (*p* = 0.007) and dementia patients (*p* = 0.025). Furthermore, the interaction term between MNA score and period (during COVID-19) was negatively associated with DBD score in dementia patients (*p* = 0.040) (Table 2). The Barthel Index scores for before vs. during COVID-19 were 85.3 ± 17.6 vs. 81.4 ± 17.7 for Daycare (*p* = 0.05), 72.5 ± 25.1 vs. 62.0 ± 18.5 for Residential (*p* = 0.129), and 85.2 ± 21.1 vs. 89.5 ± 12.8 for Visiting (*p* = 0.129) in MCI and dementia patients combined.

Analysis of government statistics showed that, by mid-March 2020, when the state of emergency was declared for COVID-19, the percentages of Daycare and Residential care service use had decreased by −1.6% and −7.7%, respectively, compared with the same month in the previous year (Figure 1). This decrease peaked in mid-May (−10.2% for Daycare and 22.2% for Residential care). Thereafter, a downward trend was observed until increases appeared in about March 2021 (of about 3% to 7% for Daycare and 3% to 10% for Residential), with an improvement trend observed from April to May 2021, one year after the peak was reached.

## 4. Discussion

We found that, during COVID-19, cognitively impaired participants with either MCI or dementia who used Daycare services, as well as those with dementia who had poor nutritional status, were significantly more likely to have worsening BPSD after adjustment for related factors.

For people with cognitive decline, several previous studies have reported the ability of different types of formal care services to prevent the onset or worsening of BPSD or alleviate the burden on caregivers [11,33,34,35,36]. Our results partially supported these previous studies, as they showed an association between a decrease in the use of Residential care services and increasing DBD score in dementia patients. Interestingly, in our model evaluating the effect of COVID-19, the use of Daycare services worsened BPSD after the start of the COVID-19 pandemic; this trend was observed in both MCI and dementia patients. In addition, in dementia patients, the use of Residential care services was borderline associated with worsening BPSD during COVID-19. Preventive measures during the COVID-19 pandemic have affected the long-term care services sector, as specifically reflected in the withholding of necessary care services and changes in the content of the care services provided [37,38,39]. In our analysis using public data, we found a trend of decreased use of Daycare and Residential care services in the first year of the COVID-19 pandemic compared with the same period in the previous year, although we observed no significant decrease in our own study. Among the care-service users we examined, Daycare users had significantly lower ADL scores after the start of COVID-19 than before, presumably reflecting the finding that those with declining physical function and worsening BPSD continued to use care services. A second possible factor behind the worsening BPSD is the change in the content of care services provided after the start of COVID-19. A survey conducted of Japanese care providers by the Japanese Care Work Foundation from December 2020 to January 2021 reported that the greatest change in the content of services provided was the “discontinuation of group events” in both Daycare and Residential care, with nearly 70% of providers curtailing events [40]. Because decreased social interaction through group events has been associated with increased dementia risk and decreased quality of life [41,42], changes in the content of care services associated with the COVID-19 pandemic could be associated with worsening BPSD. 

Additionally, a qualitative study of care services during the COVID-19 pandemic identified issues arising due to the pandemic among the care staff and families of patients [14]. Older adults receiving care services often have multiple chronic and incurable conditions including dementia, and care professionals tend to prioritize improving their QOL during care delivery. Although person-centered care was facilitated, and related symptoms such as BPSD could have been reduced when care decisions were made through in-depth communication, this study indicated a situation of inadequate communication in the pandemic environment. In summary, the COVID-19 pandemic decreased nursing care services and changed their provision to prevent infection spread among dementia patients and their family caregivers, as well as the development of infection clusters in nursing homes. As a result, the BPSD of the patients may have worsened.

Our results also revealed that good sleep quality and nutritional status had a protective effect against BPSD in both dementia and MCI patients, as did not smoking in the dementia group. Nutritional status was also significant in the model that evaluated the effects of the COVID-19 pandemic, with the finding that maintaining good nutritional status during COVID-19 was important for managing BPSD. Following is a summary of previous studies: poor nutritional status worsens BPSD, and nutritional behavior seems to worsen during the pandemic. In a Dutch epidemiologic study, the effects of the pandemic on nutritional behaviors were assessed. Those who were predisposed to overnutrition (e.g., increased snacking) reported an increase in such behavior of 20 to 32%, while those who were predisposed to undernutrition reported an increase of 7 to 15% [43]. Prior studies have also reported an association between poor nutritional status and worsening BPSD in patients with MCI or early Alzheimer’s disease [44,45]. Therefore, it can be concluded that nutritional status is more likely to be altered during the COVID-19 pandemic, leading to worsening of BPSD and thereby supporting the findings of this study, which provide new insights into the need for nutritional management in the context of the COVID-19 pandemic. Cagnin et al. reported that sleep disturbance was the most common new BPSD during COVID-19, at a rate of 21.3% in dementia patients [17]. Our previous study also showed a higher frequency of sleep disturbance during COVID-19 than in normal times [18]. We showed here that maintaining good quality of sleep could help to reduce BPSD. Sleep quality deterioration is an important early sign of mental health problems in older adults who have experienced a major change in their living environment or psychological trauma from a disaster [46,47,48]. Therefore, there is a need for a system that monitors sleep status at an early stage and leads to evidence-based sleep improvement programs that utilize, for example, pharmacotherapy and cognitive behavioral therapy [49]. In addition, tobacco dependence is a major risk factor for a number of conditions in older people, including cancer, cardiovascular disease [50], and age-related conditions such as frailty and poor work performance [51,52], as well as dementia and cognitive decline [53]. Therefore, as our findings suggest a healthy, non-smoking lifestyle may be associated with reduced BPSD. 

### Strengths and Limitations

Our study had several strengths: The inclusion of cognitively impaired patients before and during the COVID-19 pandemic provided realistic before-and-after comparisons. The results provide evidence of the need to enhance care services for patients with cognitive decline in emergency settings. However, there were some limitations to this study. One is that the design was not longitudinal: We did not follow the same subjects from before COVID-19, so it was not possible to track individual changes. In the MCI group, MMSE scores were significantly lower than those before COVID-19, and therefore, the possibility of selection bias cannot be excluded. In addition, because the study was conducted at a single institution and most of the subjects were accompanied by their family members during the assessments, we could not completely exclude information bias. Third, a previous study pointed out that BPSD varies depending on the type of dementia [17], so our study was limited by the fact that we included all-cause dementia and did not define specific dementia etiologies. In the future, we will need to analyze each type of dementia with a larger number of cases and conduct follow-up longitudinal studies, as well as comparative studies in multiple facilities with different environments, such as in urban and rural areas.

The results of this study suggest that the more serious the spread of infection in the community, the more difficult it becomes for people with dementia to receive appropriate care services, which may affect the behavioral and psychological aspects of their illness. In particular, the fact that quantitative and qualitative changes in care services are associated with worsening BPSD suggests the need to establish techniques and systems at the facility, regional, and national levels which would help provide adequate care while taking measures against infection. Additionally, the present results suggest that maintaining nutritional and other lifestyle habits may effectively prevent BPSD in patients with declining cognitive function. Based on these results, guidelines and support tools should be developed to promote measures that may prevent lifestyle changes in crises, including periods such as the COVID-19 pandemic.

## 5. Conclusions

During COVID-19, the use of Daycare services was associated with BPSD in those with cognitive decline, and poor nutritional status was also associated with BPSD in dementia patients. These findings indicate a need to examine the quantity and quality of care services for people with cognitive decline during emergencies and to continue to provide effective services.

## Figures and Tables

**Figure 1 ijerph-19-10094-f001:**
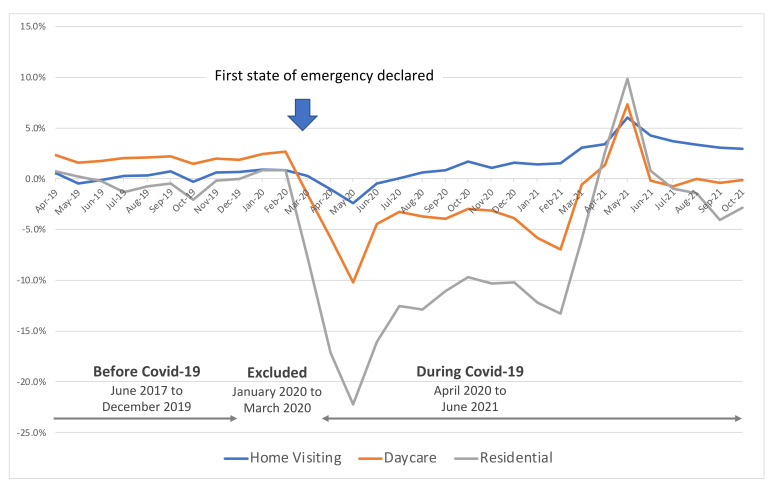
Trends in the use of care services (year-on-year change from the same month of previous year): analysis of government statistics (https://www.kokuho.or.jp/statistics/kaigo/, accessed on 24 February 2022).

**Table 1 ijerph-19-10094-t001:** Basic characteristics of participants before and during the COVID-19 pandemic.

	MCI Patients			Dementia Patients		
	Before	During	*p*-Value	Before	During	*p*-Value
	(*n* = 740)	(*n* = 237)		(*n* = 1041)	(*n* = 339)	
	*n* (%)	*n* (%)		*n* (%)	*n* (%)	
**Socioeconomic status**						
Age group						
Under 64 years	40 (5.4)	11 (4.6)	0.226	31 (3.0)	19 (5.6)	0.079
65–74 years	194 (26.2)	50 (21.1)		175 (16.8)	54 (15.9)	
75 years and over	506 (68.4)	176 (74.3)		835 (80.2)	266 (78.5)	
Gender (male)	336 (45.4)	111 (46.8)	0.757	354 (34.0)	128 (37.8)	0.233
Education (mean ± SD)	11.6 ± 4.3	11.5 ± 2.4	0.783	10.5 ± 3.2	10.7 ± 2.4	0.28
Living with family	641 (87.1)	209 (88.9)	0.528	843 (81.6)	290 (85.5)	0.115
**Use of care services**						
Home visiting	16 (2.2)	3 (1.3)	0.549	42 (4.0)	7 (2.1)	0.125
Daycare	56 (7.6)	21 (8.9)	0.614	266 (25.6)	83 (24.5)	0.748
Residential	7 (0.9)	1 (0.4)	0.715	68 (6.5)	14 (4.1)	0.135
**Medical condition**						
Clinical diagnosis						
AD	–	–	–	867 (83.3)	279 (82.3)	0.844
DLB/PD	–	–	–	114 (11.0)	41 (12.1)	
VaD	–	–	–	60 (5.8)	19 (5.6)	
Polypharmacy (5 or more)	260 (35.3)	75 (31.6)	0.344	401 (38.7)	132 (38.9)	0.981
Comorbidity (2 or more)	230 (31.1)	64 (27.0)	0.267	323 (31.0)	103 (30.4)	0.877
**Lifestyle-related variables**						
Light exercise/physical training	214 (29.2)	78 (33.1)	0.292	205 (19.9)	63 (18.7)	0.685
Quality of sleep (better)	641 (88.2)	214 (91.5)	0.203	921 (90.0)	297 (88.1)	0.376
Weight loss (N/A)	605 (82.1)	198 (84.3)	0.507	840 (81.4)	274 (81.5)	1
Non-smoking	688 (93.6)	222 (94.5)	0.747	981 (94.8)	320 (94.4)	0.891
MNA-SF (11 or above)	453 (71.5)	169 (71.6)	1	517 (57.5)	184 (54.3)	0.338
**Physical functioning**						
IADL (Lawton Index Score)						
Male (mean ± SD)	4.4 ± 1.0	4.3 ± 1.0	0.291	3.0 ± 1.4	2.9 ± 1.5	0.644
Female (mean ± SD)	7.2 ± 1.3	7.2 ± 1.2	0.991	5.0 ± 2.2	5.1 ± 2.2	0.648
Barthel index (mean ± SD)	98.2 ± 6.5	97.7 ± 7.2	0.355	92.4 ± 14.8	91.4 ± 14.5	0.284
**Cognitive functioning**						
MMSE (mean ± SD)	24.7 ± 3.3	23.8 ± 3.2	**0.001**	17.9 ± 4.9	17.8 ± 4.9	0.599

*p*-values in bold indicate statistical significance (*p* = 0.05); independent-sample *t*-test for continuous variables, and chi-squared tests for categorical variables. AD, Alzheimer’s disease; DBD, Dementia Behavioral Disturbance scale; DLB/PD, dementia with Lewy bodies and Parkinson’s disease; IADL, Instrumental Activities of Daily Living; N/A, Not Applicable; MMSE, Mini-Mental State Examination; VaD, vascular dementia; MNA, MNA-SF, Mini-Nutrition Assessment–Short Form.

**Table 2 ijerph-19-10094-t002:** Association of the Dementia Behavioral Disturbance (DBD) scale with care services and lifestyle factors in MCI and dementia patients.

	DBD			
	MCI Patients		Dementia Patients	
Variable	β (95% CI)	*p*-Value	β (95% CI)	*p*-Value
**Use of care services**				
Home visiting	−0.57 (−4.88–3.74)	0.794	2.99 (−1.06–7.05)	0.148
Daycare	−0.47 (−2.87–1.94)	0.704	−0.43 (−2.22–1.36)	0.638
Residential	−2.69 (−8.96–3.59)	0.401	−4.03 (−7.14–−0.93)	0.011
**Lifestyle-related variables**				
Light exercise/physical training	−0.60 (−1.95–0.76)	0.386	−0.64 (−2.44–1.16)	0.489
Quality of sleep (better)	−5.27 (−7.20–−3.35)	<0.001	−3.04 (−5.50–−0.57)	0.016
Weight loss (N/A)	1.01 (−0.64–2.67)	0.23	0.15 (−1.78–2.07)	0.882
Non-smoking	−0.59 (−3.07–1.89)	0.639	−4.52 (−7.62–−1.41)	0.004
MNA-SF (11 or above)	−2.19 (−3.62–−0.76)	0.003	−1.59 (−3.12–−0.07)	0.04
**Interaction terms with period (during COVID-19)**				
Home visiting	0.46 (−11.21–12.12)	0.938	−4.45 (−14.72–5.82)	0.395
Daycare	5.80 (1.59–10.01)	0.007	3.66 (0.47–6.85)	0.025
Residential	−2.95 (−19.23–13.34)	0.723	6.39 (−0.15–12.94)	0.056
Light exercise/physical training	−0.78 (−3.26–1.70)	0.535	−0.04 (−3.54–3.45)	0.981
Quality of sleep (better)	−0.16 (−4.25–3.93)	0.937	−3.68 (−8.16–0.80)	0.107
Weight loss	0.12 (−3.11–3.35)	0.943	−0.97 (−4.61–2.67)	0.601
Non-smoking	−1.46 (−6.40–3.47)	0.561	4.21 (−1.59–10.01)	0.155
MNA-SF (11 or above)	−0.33 (−3.03–2.37)	0.811	−1.65 (−3.12–−0.07)	0.04

Logistic regression models adjusted for time period, age, gender, education, living with family, polypharmacy, Mini-Mental State Examination, and Instrumental Activities of Daily Living. Additionally, type of dementia was entered into the dementia patient group. CI, confidence interval; MCI, mild cognitive impairment; MNA-SF, Mini-Nutrition Assessment–Short Form.

## Data Availability

The data presented in this study are available on request from the corresponding author. The data are not publicly available due to their containing information that could compromise the privacy of research participants.

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
