# Peer review of "Factors Associated with Behavioral and Psychological Symptoms of Dementia during COVID-19"

_ijerph, 2022, doi:10.3390/ijerph191610094_

Round 1
Reviewer 1 Report
This study analyzed the changes in the use of care services of MCI and dementia patients before and during COVID-19, and the effect on BPSD along with lifestyle.
Since the outcome variable of this study is BPSD, the authors need to compare the frequency of each sub-item of BPSD measured by DBD before and after COVID-19. DBD consists of 28 observable behaviors related to dementia, such as passivity, agitation, eating disturbances, aggressiveness, diurnal rhythm disturbances, and sexual disinhibition.
Table 1 of the following reference paper No. 16 and Table 1 of this paper are almost identical.
16. Kuroda Y, Sugimoto T, Matsumoto N, Uchida K, Kishino Y, Suemoto CK, et al. Prevalence of Behavioral and Psychological 324 Symptoms in Patients With Cognitive Decline Before and During the COVID-19 Pandemic. Frontiers in Psychiatry [Internet]. 325 2022 [cited 2022 Mar 25];13. Available from: https://www.frontiersin.org/article/10.3389/fpsyt.2022.839683
It is necessary to further describe the differences between the data set and the composition and content of this study and the reference paper No. 16.
The study findings can be the evidence for including contents of quality of sleep and nutritional status in programs to improve or prevent BPSD.
It will be helpful for readers to understand the research results, if the period after diagnosis of dementia is included in Table 1.
The validity of this paper will increase if more improved contents from the paper linked below in SSRN are included in this paper.
https://papers.ssrn.com/sol3/papers.cfm?abstract_id=4109740
Author Response
Reviewer #1
This study analyzed the changes in the use of care services of MCI and dementia patients before and during COVID-19, and the effect on BPSD along with lifestyle.
Comment 1.
Since the outcome variable of this study is BPSD, the authors need to compare the frequency of each sub-item of BPSD measured by DBD before and after COVID-19. DBD consists of 28 observable behaviors related to dementia, such as passivity, agitation, eating disturbances, aggressiveness, diurnal rhythm disturbances, and sexual disinhibition.
Response
Thank you for this valuable comment. As the reviewer pointed out, the DBD consists of 28 items, each of which assesses the degree of BPSD. Our previous study, discussed in the following comments, showed the prevalence of each DBD items. The significant difference between this study and our previous study is that it focused on factors related to BPSD and used the total BPSD scores in its analysis. On the other hand, as the reviewer pointed out, it would better assist the reader's understanding of the prevalence of each item shown, so we prepared a Supplemental Table.
Comment 2.
Table 1 of the following reference paper No. 16 and Table 1 of this paper are almost identical.
- Kuroda Y, Sugimoto T, Matsumoto N, Uchida K, Kishino Y, Suemoto CK, et al. Prevalence of Behavioral and Psychological 324 Symptoms in Patients With Cognitive Decline Before and During the COVID-19 Pandemic. Frontiers in Psychiatry [Internet]. 325 2022 [cited 2022 Mar 25];13. Available from: https://www.frontiersin.org/article/10.3389/fpsyt.2022.839683
It is necessary to further describe the differences between the data set and the composition and content of this study and the reference paper No. 16.
Response
Thank you for this perspective. As pointed out by reviewer, we added following explanation: we clearly stated that the paper cited in #18 was our previous study and clarified the difference between the data set used in that previous study and the data set in this study.
L80-83
“Therefore, in our previous study, we compared the prevalence of BPSD using continuous data from 15 months before and during COVID-19, and found that sleep disturbances and aggressiveness were more prevalent after the start of COVID-19.[18]”
L108-111
“The data set used in this study was designed to be highly extensive before the COVID-19 period compared to the dataset used in our previous study[18], resulting in an increased number of participants. Independent researchers ensured that the same participants was not evaluated at the two time points.”
Comment 3.
The study findings can be the evidence for including contents of quality of sleep and nutritional status in programs to improve or prevent BPSD.
Response
Thank you for this important comment. Our next steps will be to conduct a longitudinal study to gain more robust findings and link them to support programs to alleviate BPSD and related symptoms.
Comment 4.
It will be helpful for readers to understand the research results, if the period after diagnosis of dementia is included in Table 1.
Response
Thank you for this valuable comment. As the reviewer pointed out, including the period after the diagnosis of dementia would be more helpful in understanding the results; however, since we included only first-time patients, information on the specific period between the diagnosis and assessment of dementia was not available. In future longitudinal studies, we would like to obtain information on the above period by confirming with the referring medical institution.
Comment 5.
The validity of this paper will increase if more improved contents from the paper linked below in SSRN are included in this paper.
https://papers.ssrn.com/sol3/papers.cfm?abstract_id=4109740
Response
Thank you for your helpful comments; the paper submitted to SSRN is from a previous submission to another journal. We have improved this paper significantly by adding the background and implications of the study based on the reviewer's comments.

Reviewer 2 Report
The present study investigated which factors contributed to the worsening of BPSD during COVID-19 pandemic in MCI and dementia patients of a memory clinic in Japan. The authors conclude that the use of daycare services and poor nutritional status was associated with BPSD worsening, by comparing the information of patients attending the memory clinic in two different period (June 2017-December 2019 and April 2020-June 2021).
Overall, the manuscript is clear and well-structured and the topic focused for the special issue.
I only have some comments and suggestions to improve the description of the study context and design.
1. I think it could be helpful to add some details on the impact of COVID-19 on dementia care services in the specific context in which data were collected, since each nation adopted different measurements to counteract the COVID-19 spread. Is there any period in which the access to the services were reduced, interrupted, or access criteria changed? I suggest to briefly describe it in the Introduction, also mentioning in which periods the measurements were set. Please also consider to add further interpretations in the discussion on how these aspects eventually affected the results. In the present form, the discussion only included some considerations on the change of the content of care services during the pandemic based on a survey conducted of Japanese care providers.
2. The major limitation of the present study is to adopt a transversal design instead of a longitudinal one, which would be more appropriate for the aim of studying the factors affecting BPSD worsening in different periods. This aspect is explicitly stated only in the strengths and limitations section. I think it could be useful to better describe the study design early in the manuscript. For example, I suggest to add in the "Study cohort" paragraph more details on the inclusion criteria. Moreover, it is not clear how multiple evaluations of the same patients within the study period are considered, which are usually performed in memory clinic contexts to assess disease progression. Do the authors included only the first assessment visit for each patient? As regards the comparisons of different group of patients for different period, do the authors think that there is any "selection bias" of patients attending the memory clinic during the COVID-19 period which may affect the results? For example, Table 1 shows a significant difference in cognitive functioning (MMSE score) before and during the pandemic in MCI patients.
3. Table 1 shows that during the pandemic the number of care services users in MCI patients subgroup is extremely low. I don't think that the analysis performed is adequately powered to model an interaction between use of care services and period (Table 2). Please provide explanations for it.
4. The significance and implications of the findings should be better argued in the discussion. Why daycare services use and nutritional status affect BPSD worsening during the COVID-19? How this findings may help dementia care providers in their work?
Other minor revisions:
Page 3, Explanatory variables paragraph: please details the responses corresponding to the 4-point scale of the lifestyle questionnaire, as for the other scales described.
Page 4, Results section: Please report in the text or in the table the statistical parameters of the analysis showing that the main effect of the COVID-19 period was not significant (beginning of the second paragraph).
There are some typos in writing COVID-19 (sometimes all capital letters sometimes not, 2 occurrences of "COVIC-19" in the statistical analysis and results paragraph respectively).
Author Response
Reviewer #2
The present study investigated which factors contributed to the worsening of BPSD during COVID-19 pandemic in MCI and dementia patients of a memory clinic in Japan. The authors conclude that the use of daycare services and poor nutritional status was associated with BPSD worsening, by comparing the information of patients attending the memory clinic in two different period (June 2017-December 2019 and April 2020-June 2021).
Overall, the manuscript is clear and well-structured and the topic focused for the special issue.
I only have some comments and suggestions to improve the description of the study context and design.
Comment 1.
I think it could be helpful to add some details on the impact of COVID-19 on dementia care services in the specific context in which data were collected, since each nation adopted different measurements to counteract the COVID-19 spread. Is there any period in which the access to the services were reduced, interrupted, or access criteria changed? I suggest to briefly describe it in the Introduction, also mentioning in which periods the measurements were set. Please also consider to add further interpretations in the discussion on how these aspects eventually affected the results. In the present form, the discussion only included some considerations on the change of the content of care services during the pandemic based on a survey conducted of Japanese care providers.
Response
We would like to express our deep appreciation for the reviewers' careful review and valuable comments. As the reviewer pointed out, countries have taken different measures to control COVID-19 dissemination. Therefore, a paragraph detailing the impact of care services at the time the data was collected has been included in the Introduction section.
>L60-70
“However, with the outbreak of the COVID-19 pandemic, precautionary measures were taken to limit the number of gatherings in Japan, and the use of outpatient and long-term care services significantly reduced throughout the country[13]. Further, the services provided in long-term care facilities were also affected by the precautionary measures, which included social distancing and other infection-control measures[14]. However, despite these preventive measures, COVID-19 infection clusters still formed in nursing homes and services had to be suspended during specific periods. Thus, the reduction or suspension of services due to the pandemic may have worsened the dementia symptoms of patients, consequently increasing caregiving burden, and psychological and physical health problems among care professionals.”
Also, in our discussion of this paper, we stated, based on the questionnaire survey, that the decreased quality and quantity of care services during COVID-19 was associated with worsening BPSD, but we further deepened our discussion by citing qualitative studies that support our argument.
>L254-264
“Additionally, a qualitative study of care services during the COVID-19 pandemic identified issues arising due to the pandemic among the care staff and families of patients [14]. Older adults receiving care services often have multiple chronic and non-curative conditions including dementia, and care professionals tend to prioritize improving their QOL during care delivery. Although person-centered care was facilitated and related symptoms such as BPSD could have reduced when care decisions were formed through in-depth communication, this study indicated a situation of inadequate communication in the pandemic environment. In summary, the COVID-19 pandemic decreased nursing care services and changed their provision to prevent infection spread among dementia patients and their family caregivers, as well as the development of infection clusters in nursing homes. As a result, the BPSD of the patients may have worsened.”
Comment 2.
The major limitation of the present study is to adopt a transversal design instead of a longitudinal one, which would be more appropriate for the aim of studying the factors affecting BPSD worsening in different periods. This aspect is explicitly stated only in the strengths and limitations section. I think it could be useful to better describe the study design early in the manuscript. For example, I suggest to add in the "Study cohort" paragraph more details on the inclusion criteria. Moreover, it is not clear how multiple evaluations of the same patients within the study period are considered, which are usually performed in memory clinic contexts to assess disease progression. Do the authors included only the first assessment visit for each patient? As regards the comparisons of different group of patients for different period, do the authors think that there is any "selection bias" of patients attending the memory clinic during the COVID-19 period which may affect the results? For example, Table 1 shows a significant difference in cognitive functioning (MMSE score) before and during the pandemic in MCI patients.
Response
Thank you for these important perspectives. As the reviewer pointed out, we recognized the need to explain the study design in more detail earlier in the manuscript, so we added the following text:
L94-96
“The aim of this study was to analyze BPSD factors during two different periods, before and during the COVID-19 pandemic. Therefore, a cross-sectional study was conducted at two different time points.”
We also explained that we did not include the same patients in two different periods.
L110-111
“Independent researchers ensured that the same participants was not evaluated at the two time points.”
Furthermore, as the reviewer pointed out, the possibility of selection bias cannot be completely excluded, so we have added the following sentence to the study limitations.
L300-302
“In the MCI group, MMSE scores were significantly lower than those before COVID-19, and therefore, the possibility of selection bias cannot be excluded.”
Comment 3.
Table 1 shows that during the pandemic the number of care services users in MCI patients subgroup is extremely low. I don't think that the analysis performed is adequately powered to model an interaction between use of care services and period (Table 2). Please provide explanations for it.
Response
Thank you for pointing out to us. As the reviewer indicated, the number of long-term care service users in the MCI group was small. Additionally, the interaction between the use of care services and period was not significant. We consulted our co-author statistician again, who responded as follows:
As the reviewer indicated, it is possible that the true effect of the interaction term was small or that the low frequency resulted in a low power. However, it is generally challenging to check for statistical power, making it impossible to determine the reason(s) for this lack of significance.
However, the regression coefficients and their standard errors were estimated without any problem. In other words, the frequency of the observations was sufficient to perform a linear multiple regression model; however, the results were not significant. Confidence intervals are shown in Table 2, which shows that the regression coefficients and standard errors are estimated without problem.
Comment 4.
The significance and implications of the findings should be better argued in the discussion. Why daycare services use and nutritional status affect BPSD worsening during the COVID-19? How this findings may help dementia care providers in their work?
Response
Thank you for these thoughts. As the reviewer pointed out, the importance of the findings and their implications were not adequately discussed. Therefore, the following discussion has been added:
L254-264
“Additionally, a qualitative study of care services during the COVID-19 pandemic identified issues arising due to the pandemic among the care staff and families of patients [14]. Older adults receiving care services often have multiple chronic and non-curative conditions including dementia, and care professionals tend to prioritize improving their QOL during care delivery. Although person-centered care was facilitated and related symptoms such as BPSD could have reduced when care decisions were formed through in-depth communication, this study indicated a situation of inadequate communication in the pandemic environment. In summary, the COVID-19 pandemic decreased nursing care services and changed their provision to prevent infection spread among dementia patients and their family caregivers, as well as the development of infection clusters in nursing homes. As a result, the BPSD of the patients may have worsened.”
L260-280
“Following is the summary of previous studies: poor nutritional status worsens BPSD and nutritional behavior seems to get worse during the pandemic. In a Dutch epidemiologic study, effects of the pandemic on nutritional behaviors were assessed. Those who were predisposed to overnutrition (e.g., increased snacking) reported an increase in such behavior by 20 to 32%, while those who were predisposed to undernutrition reported an increase by 7 to 15%.[43] Prior studies have also reported an association between poor nutritional status and worsening BPSD in patients with MCI or early Alzheimer’s disease. [44,45] Therefore, it can be concluded that nutritional status is more likely to be altered during the COVID-19 pandemic, leading to worsening of the BPSD, and thereby supporting the findings of this study, which provide new insights into the need for nutritional management in the context of the COVID-19 pandemic.”
L311-321
“The results of this study suggest that the more serious the spread of infection in the community, the more difficult it becomes for people with dementia to receive appropriate care services, which may affect the behavioral and psychological aspects of their illness. In particular, the fact that quantitative and qualitative changes in care services are associated with worsening BPSD suggests the need to establish techniques and systems at facility, regional, and national levels, which would help provide adequate care while taking measures against the infection. Additionally, the present results suggest that maintaining nutritional and other lifestyle habits may effectively prevent BPSD in patients with declining cognitive function. Based on these results, guidelines and support tools should be developed to promote measures that may prevent lifestyle changes in crises, including periods such as the COVID-19 pandemic”
Comment 5.
Other minor revisions:
Page 3, Explanatory variables paragraph: please details the responses corresponding to the 4-point scale of the lifestyle questionnaire, as for the other scales described.
Response
In accordance with the reviewer's remarks, the following explanation has been added.
L132-135
“For the “light exercise/physical training” and “sleep quality” variables, a “good” status was counted as 3 or 4 on a 4-point scale, whereas weight loss was counted as “not available (N/A),” and smoking was counted as “no smoking”.”
Comment 6.
Page 4, Results section: Please report in the text or in the table the statistical parameters of the analysis showing that the main effect of the COVID-19 period was not significant (beginning of the second paragraph).
Response
In accordance with the reviewer's remarks, the following statistical parameters has been added.
L179-180
“(MCI group β = 2.12, CI -4.39 to 8.63; Dementia group β = 1.45, CI -5.74 to 8.64).”
Comment 7.
There are some typos in writing COVID-19 (sometimes all capital letters sometimes not, 2 occurrences of "COVIC-19" in the statistical analysis and results paragraph respectively).
Response
Thank you for this input. We have reviewed the terminology throughout the paper in accordance with the reviewers' suggestions.
